# Inverse reconstruction of model cells: Extracting structural and molecular insights through infrared spectroscopic cytology

**Proity Nayeeb Akbar** [ID]◉*, **Reinhold Blümel**◉

Department of Physics, Wesleyan University, Middletown, Connecticut, United States of America

◉ These authors contributed equally to this work.

* pakbar@wesleyan.edu

## Abstract

Infrared (IR) microspectroscopy stands as a transformative clinical tool for analyzing single biological cells in biopsy samples, offering critical insights into their chemical composition. In this study, we further develop a recently proposed inverse scattering algorithm that accurately reconstructs the dielectric properties of single cells, considering both scattering and absorption. We demonstrate the method's effectiveness using spherical model cells filled with six organic test substances: polymethyl methacrylate (PMMA), polycarbonate (PC), polydimethylsiloxane (PDMS), polyetherimide (PEI), polyethylene terephthalate (PET), and polystyrene (PS). The permittivity values of these substances, reconstructed from their extinction efficiencies and known refractive indexes from the literature, show excellent agreement with experimental data. Our comparative analysis of the basis sets for the reconstruction algorithm reveals that using dielectric functions leads to more accurate results compared to anti-symmetrized Lorentzians. We find that compared to other methods in the literature on PMMA spheres, our approach yields reconstructions of significantly higher quality. These findings not only enhance reconstruction accuracy but also advance the potential of IR microspectroscopy for clinical cytology, where precise molecular analysis is crucial for disease diagnosis and monitoring at the cellular level.

## 1 Introduction

The study of single cells in medical research is crucial for investigating biological processes altered by heterogeneities arising in large populations of cells. Cells are dynamic, changing in time through cell differentiation depending on microenvironmental factors [1]. Even the tiniest perturbations (internal or external) to cell signaling can promote the onset of disease, like cancer, that may best be treated if diagnosed early on at the initial stages of cell proliferation [2].

**Data availability statement:** All data that support the findings of this study are included within the article.

**Funding:** The author(s) received no specific funding for this work.

**Competing interests:** The authors have declared that no competing interests exist.

In principle, synchrotron infrared (IR) microspectroscopy is a powerful clinical tool to analyze single biological cells in biopsy samples at subcellular resolution and to detect abnormal cellular activity [3–8]. Applying synchrotron radiation, however, is expensive, directing most clinical environments to opt for commercial bench-top infrared instrumentation, which lacks the high-end spatial resolution offered by synchrotron-based techniques [8–11]. Additionally, because of the heterogeneity of biological materials [12], and spectroscopic complications due to the small sample size, which leads to significant scattering effects, the data obtained is often spurious [13,14]. Thus, improved spectroscopic analysis is cardinal in building a complete understanding of the sample's morphology.

To circumvent these difficulties, we take a wave-optics-based approach, in which an obtained IR spectrum is a combination of effects such as absorption of radiation by the molecules, signals from scattering, and other optical phenomena that depend on the geometric cross-section of the sample and the wave nature of the infrared radiation [15–17]. If we want to determine the internal chemical structure of cells, we have to ensure that the scattering contribution to the measured absorption is eliminated so that only the pure absorption due to the functional groups is retrieved, which carries all the necessary information about the chemical features of the cell [18].

Therefore, we propose that recovering the space- and frequency-resolved absorption of a cell, i.e., its hyperspectral absorption map, will allow for reconstructing the cell altogether. Our rationale is that the construction of the hyperspectral absorption map, i.e., the fingerprint of the cellular functional groups, will help identify any abnormalities at the molecular level and offer new diagnostic [19], prognostic [20], and treatment opportunities for medical conditions [21], including cancer. Consequently, since the hyperspectral absorption map is encoded in the space- and frequency-dependent imaginary part of the permittivity of a cell, the objective of this research is to find ways to extract the cellular space- and frequency-dependent complex permittivity, encoded in the scattering and absorption properties of a cell, by solving an inverse scattering problem.

Several methods exist for removing instrumental noise and unwanted artifacts, such as scattering effects, from IR spectra, but each comes with its own set of limitations. For example, signal reconstruction using total variation regularization is effective in preserving sharp edges in the IR spectra, but it can also result in the loss of fine details [22]. There are deep-learning methods [23,24] that offer promising solutions for correcting band distortions, but they require large amounts of prior knowledge and spectral data for training, making them less applicable in scenarios with limited data. There are also powerful methods based on extended multiplicative signal correction [25,26] for which an open-source code is published [18]. In contrast to this method, however, which requires a reference spectrum to get started, our method, to be described and tested in this paper, is a direct, inverse-scattering method that does not require a reference spectrum and thus does not introduce a possible bias into the spectral reconstruction.

Macromolecules, i.e., the long polymers that construct biological cells [27], are essential in understanding IR absorption spectra for cell cytology [28]. In physical terms, aggregates of these complex molecules that make up a cell form dielectric materials [29], generating electromagnetic responses due to the various functional groups of the macromolecules as they interact with specific IR frequencies to generate the absorption bands we observe [30]. Understanding the dielectric response of polymeric materials can provide insight into the quantitative and qualitative behavior of proteins, lipids, carbohydrates, and nucleic acids as they share many common functional groups [27]. In a previous study [31], we presented a novel method for the extraction of the complex refractive index from the extinction efficiency of homogeneous spheres composed of polymethylmethacrylate and polystyrene, using anti-symmetrized

Lorentzian functions as the basis set for the reconstruction algorithm. We found that a non-linear optimization algorithm was able to extract chemical information via direct reconstruction of the space- and frequency-dependent complex refractive index. Despite being reliable and accurate, there is a basis set that is even more suitable, i.e., the set of dielectric functions, which we discuss in more detail in the following sections. In addition to exploring the performance of the set of dielectric functions, this paper also explores the extraction of the signatures of the functional groups from extinction spectra of a larger number of test materials, i.e., polymethylmethacrylate, polycarbonate, polydimethylsiloxane, polyetherimide, polyethylene terephthalate, and polystyrene. Extending the number and types of test polymers is important for two primary reasons, i.e., **i)** to verify that our algorithm is robust and is expected to work for any organic material and **ii)** the chosen polymers share many of the same functional groups encountered in biological materials.

Our results are significant as they not only facilitate the detection of malignancies in biological cells and thus have the potential to elucidate tumor grades, but also motivate the large-scale application of bench-top infrared microspectroscopy in the health community. Furthermore, this approach does not require prior knowledge of spectroscopic principles, which ensures accurate data interpretation and, in turn, improves the predictive value of current IR cytological methods. A firm understanding of the unusual spectral effects observed in cells resulting from the heterogeneity of the cellular spectra can then help in the development of IR microspectroscopy as a diagnostic indicator of disease.

The paper is structured as follows. In Section 2 we present the methods that centrally underlie our study. In Section 3 we present our results, demonstrating the power of our reconstruction algorithm in determining the complex permittivities (the complex refractive indexes) from given extinction spectra of six polymer test substances. In Section 4 we summarize our results and conclude the paper.

In the IR region, spectroscopic information is usually presented as a function of wavenumber in units of cm$^{-1}$ or as a function of wavelength in units of $\mu$m. Throughout this paper our theory is formulated in terms of wavenumber. For the convenience of the reader, spectroscopic data in the text are stated "bilingually" both in wavenumbers and associated wavelengths (in parentheses). In addition, all figures carry two scales, a bottom scale in wavenumbers and a top scale in wavelengths.

## 2 Methods

We start by describing the theory behind IR spectroscopy (Section 2.1) and the classical Lorentz oscillator model (Section 2.2) that explains the dielectric behavior of the system. We follow it up with a brief description of the Kramers-Kronig transformation (Section 2.3), which relates the real and imaginary parts of the complex dielectric functions to each other. In Sections 2.4 and 2.5, we define our inverse scattering algorithm using a wave-optics based approach, and describe our method for reconstructing the frequency-dependent complex permittivity (the complex refractive index, respectively). In Section 2.6, we elaborate on the materials used and the importance of testing a range of different functional groups to validate the robustness of the reconstruction algorithm for the extraction of chemical and biological properties from single cells.

### 2.1 General IR spectroscopy for cell imaging

As one of the most powerful, non-destructive, and label-free tools in the analysis of tissues, IR spectroscopy has long been used in medical research to study molecular reactions in individual living cells [28,32,33]. It exploits the electromagnetic spectrum's mid-infrared region

to determine the physical properties of cell components and to identify the chemical composition of compounds originating from the characteristic vibrational frequencies of functional groups [34,35]. We define the wavenumber

$$\tilde{\nu} = 1/\lambda, \tag{1}$$

where $\lambda$ is the vacuum wavelength of the IR radiation. Single-detector IR spectroscopy proceeds by illuminating a biological cell with a beam of IR radiation of a given wavenumber $\tilde{\nu}$ and intensity $I_0(\tilde{\nu})$, and measuring the transmitted intensity, $I(\tilde{\nu})$, in the forward direction. The result of the measurement is usually stated as the measured absorbance, also called the apparent absorbance, defined as

$$\mathcal{A}(\tilde{\nu}) = -\log_{10}\left[\frac{I(\tilde{\nu})}{I_0(\tilde{\nu})}\right]. \tag{2}$$

Given the reception area, $G$, of the detector, the geometric cross-section, $g$, of the sample ($G \gg g$), and the extinction cross-section [36], $\sigma_{\text{ext}}$, we represent $\mathcal{A}(\tilde{\nu})$ in terms of the extinction efficiency [12]

$$Q_{\text{ext}}(\tilde{\nu}) = \frac{\sigma_{\text{ext}}(\tilde{\nu})}{g} = \frac{GI_0(\tilde{\nu}) - GI(\tilde{\nu})}{gI_0(\tilde{\nu})} = \left(\frac{G}{g}\right)\left[1 - \left(\frac{I(\tilde{\nu})}{I_0(\tilde{\nu})}\right)\right] = \left(\frac{G}{g}\right)\left[1 - 10^{-\mathcal{A}(\tilde{\nu})}\right]. \tag{3}$$

Since, via Eq (3), $Q_{\text{ext}}(\tilde{\nu})$ and $\mathcal{A}(\tilde{\nu})$ contain the same information, it is possible to perform reconstructions of the permittivities based on either $Q_{\text{ext}}(\tilde{\nu})$ or $\mathcal{A}(\tilde{\nu})$. For the purposes of this paper, we perform our reconstructions exclusively on the basis of $Q_{\text{ext}}(\tilde{\nu})$.

## 2.2 Lorentz oscillator model for electromagnetic response of cells

Biological cells are composed of layers and inclusions of materials with different complex permittivities [29]. These differences in dielectric composition come about as a consequence of the varying functional groups inside the different parts of a cell. The question of identifying the functional groups of a cell then reduces to understanding the dielectric properties of the biological material. Consequently, we use the classical Lorentz Oscillator model [31,37–39] to describe the response of biological cells, modeled as dielectric materials, to an electromagnetic wave.

The Lorentz oscillator model describes an electron as a driven, damped harmonic oscillator and assumes that the electron is connected to the nucleus via a harmonic force with spring constant $k$. The driving force is the oscillating electric field. The damping force models dissipative effects of the electron that oppose the driving electric field and ensures that the amplitudes of the oscillations do not diverge to infinity when the driving frequency approaches the resonance frequency. The model then uses Newton's second law to obtain the electronic equation of motion

$$m_e \ddot{\vec{r}} + m_e \Gamma \dot{\vec{r}} + m_e \omega_0^2 \vec{r} = -e\vec{E}, \tag{4}$$

where $m_e$ and $-e$ are the mass and charge of the electron, respectively, $\vec{r}$ is the displacement vector, $\Gamma$ is the damping constant, $\dot{\vec{r}}$ is the velocity, $\omega_0 = \sqrt{k/m_e}$ is the natural oscillator frequency, and $\vec{E}$ is the applied electric field. Taking the Fourier transform of Eq (4) and simplifying and rearranging, we obtain the complex amplitude of the electron's motion

$$\vec{r}(\tilde{\nu}) = \left( \frac{-e}{m_e (2\pi c)^2} \right) \left( \frac{\vec{E}(\tilde{\nu})}{\tilde{\nu}_0^2 - \tilde{\nu}^2 - i\tilde{\nu}\gamma} \right), \tag{5}$$

where $\tilde{\nu} = \omega/(2\pi c)$, $\gamma = \Gamma/(2\pi c)$, and $c$ is the vacuum speed of light. Using the definition of the electric dipole moment,

$$\vec{\mu}(\tilde{\nu}) = -e\vec{r}(\tilde{\nu}), \tag{6}$$

along with the volume average of the electric dipole moment over the total number of atoms per unit volume, $N$, we obtain the following relationship for the polarization $\vec{P}(\tilde{\nu})$ per unit volume:

$$\vec{P}(\tilde{\nu}) = < N\vec{\mu}(\tilde{\nu}) > = \frac{Ne^2}{m_e (2\pi c)^2} \left( \frac{\vec{E}(\tilde{\nu})}{\tilde{\nu}_0^2 - \tilde{\nu}^2 - i\tilde{\nu}\gamma} \right). \tag{7}$$

The complex linear polarization $\vec{P}(\tilde{\nu})$ is related to the complex electric susceptibility, $\tilde{\chi}_e(\tilde{\nu})$, according to

$$\vec{P}(\tilde{\nu}) = \epsilon_0 \tilde{\chi}_e(\tilde{\nu}) \vec{E}(\tilde{\nu}), \tag{8}$$

where $\epsilon_0$ is the permittivity of the vacuum. Thus, Eq (7) and Eq (8) yield the complex permittivity, $\tilde{\epsilon}_r(\tilde{\nu})$, according to

$$\tilde{\epsilon}_r(\tilde{\nu}) = 1 + \tilde{\chi}_e(\tilde{\nu}) = 1 + \frac{\tilde{\nu}_p^2}{\tilde{\nu}_0^2 - \tilde{\nu}^2 - i\tilde{\nu}\gamma}, \tag{9}$$

where $\tilde{\nu}_p^2 = Ne^2/[\epsilon_0 m_e (2\pi c)^2]$. The complex permittivity, $\tilde{\epsilon}_r(\nu)$, can be split into its real and imaginary components,

$$\tilde{\epsilon}_r'(\tilde{\nu}) = \epsilon_\infty + \frac{\tilde{\nu}_p^2 \left( \tilde{\nu}_0^2 - \tilde{\nu}^2 \right)}{\left( \tilde{\nu}_0^2 - \tilde{\nu}^2 \right)^2 + \left( \tilde{\nu}\gamma \right)^2}, \tag{10}$$

$$\tilde{\epsilon}_r''(\tilde{\nu}) = \frac{\tilde{\nu}\gamma \tilde{\nu}_p^2}{\left( \tilde{\nu}_0^2 - \tilde{\nu}^2 \right)^2 + \left( \tilde{\nu}\gamma \right)^2}, \tag{11}$$

respectively, where $\epsilon_\infty$, replacing "1" in Eq (9), is an empirically (experimentally) determined constant offset that accounts for deviations of the complex permittivity from unity due to the contribution of absorption bands at frequencies away from the resonance frequency of the oscillator. With the general form of the complex refractive index, i.e.,

$$\tilde{n}^2(\tilde{\nu}) = \tilde{\epsilon}_r(\tilde{\nu}), \tag{12}$$

we express the complex refractive index in terms of its real, $\tilde{n}'(\tilde{\nu})$, and imaginary, $\tilde{n}''(\tilde{\nu})$, components

$$\tilde{n}(\tilde{\nu}) = \tilde{n}'(\tilde{\nu}) + i\tilde{n}''(\tilde{\nu}), \tag{13}$$

where, according to the Lorentz Oscillator model, the analytical forms of the real and imaginary parts of $\tilde{n}(\tilde{\nu})$ are given explicitly by

$$\tilde{n}'(\tilde{\nu}) = \sqrt{\frac{|\tilde{\varepsilon}_r(\tilde{\nu})| + \tilde{\varepsilon}_r'(\tilde{\nu})}{2}}, \tag{14}$$

$$\tilde{n}''(\tilde{\nu}) = \sqrt{\frac{|\tilde{\varepsilon}_r(\tilde{\nu})| - \tilde{\varepsilon}_r'(\tilde{\nu})}{2}}, \tag{15}$$

respectively, where $|\tilde{\varepsilon}_r(\tilde{\nu})|$ is the modulus of the complex dielectric permittivity. Real and imaginary parts of $\tilde{\varepsilon}_r$ and $\tilde{n}$ are connected via the following relationships:

$$\tilde{\varepsilon}_r'(\tilde{\nu}) = \tilde{n}'(\tilde{\nu})^2 - \tilde{n}''(\tilde{\nu})^2, \tag{16}$$

$$\tilde{\varepsilon}_r''(\tilde{\nu}) = 2\tilde{n}'(\tilde{\nu})\tilde{n}''(\tilde{\nu}). \tag{17}$$

Assuming that the dielectric function in Eq (11) is sharply peaked at $\tilde{\nu} = \tilde{\nu}_0$, we may write approximately [40],

$$\tilde{\varepsilon}_r''(\tilde{\nu}) \approx \frac{\tilde{\nu}_0\gamma\tilde{\nu}_p^2}{\left(\tilde{\nu}_0^2 - \tilde{\nu}^2\right)^2 + \left(\tilde{\nu}_0\gamma\right)^2} \approx \frac{\tilde{\nu}_0\gamma\tilde{\nu}_p^2}{\left(\tilde{\nu}_0 - \tilde{\nu}\right)^2 4\tilde{\nu}_0^2 + \left(\tilde{\nu}_0\gamma\right)^2} = \frac{\gamma\tilde{\nu}_p^2/(4\tilde{\nu}_0)}{\left(\tilde{\nu}_0 - \tilde{\nu}\right)^2 + (\gamma/2)^2}. \tag{18}$$

This shows that in the vicinity of an absorption band, $\tilde{\varepsilon}_r''(\tilde{\nu})$ may be approximated by a Lorentzian function [40], which formed the foundation of our procedures in our previous work [31]. In the current paper, however, we use the correct dielectric functions, Eqs (10) and (11), to reconstruct complex permittivities from extinction spectra. Since the literature typically presents experimental refractive index data for materials rather than their complex permittivities, any reference to experimental permittivities in this paper refers to those computed by converting the experimental refractive indexes to permittivities using Eqs (16) and (17), respectively.

While the permittivities $\tilde{\varepsilon}_r(\tilde{\nu})$ defined in Eqs (10) and (11) determine the extinction spectrum $Q_{\text{ext}}(\tilde{\nu})$ defined in Eq (3), no simple analytical formula exists that would yield $Q_{\text{ext}}(\tilde{\nu})$ as a function of $\tilde{\varepsilon}_r(\tilde{\nu})$. Nevertheless, powerful standard numerical codes exist that perform this functional relationship effectively and accurately [39]. The reverse direction, i.e., determining $\tilde{\varepsilon}_r(\tilde{\nu})$ from $Q_{\text{ext}}(\tilde{\nu})$ is much more difficult to perform, and no standard codes exist. Thus, the relationship $\tilde{\varepsilon}_r(\tilde{\nu}) \leftrightarrow Q_{\text{ext}}(\tilde{\nu})$ is akin to a trapdoor function in mathematical cryptography [41], where one direction [in our case $\tilde{\varepsilon}_r(\tilde{\nu}) \to Q_{\text{ext}}(\tilde{\nu})$] is easy to perform, while the reverse direction [in our case $\tilde{\varepsilon}_r(\tilde{\nu}) \leftarrow Q_{\text{ext}}(\tilde{\nu})$] is exceedingly hard to perform. Presenting an efficient and accurate method for performing the reverse direction $\tilde{\varepsilon}_r(\tilde{\nu}) \leftarrow Q_{\text{ext}}(\tilde{\nu})$, i.e., reconstructing $\tilde{\varepsilon}_r(\tilde{\nu})$ from $Q_{\text{ext}}(\tilde{\nu})$, is the central topic of our paper.

## 2.3 Kramers-Kronig transformation

It is straightforward to verify by explicit computation that the real and imaginary parts, Eqs (10) and (11), respectively, of the complex permittivity $\tilde{\varepsilon}_r(\tilde{\nu})$, derived from the Lorentz Oscillator model, satisfy the Kramers-Kronig relations

$$\tilde{\varepsilon}_r'(\tilde{\nu}) = \varepsilon_\infty + \left(\frac{2}{\pi}\right)\mathcal{P}\int_0^\infty \left[\frac{\Omega\tilde{\varepsilon}_r''(\Omega)}{\tilde{\nu}^2 - \Omega^2}\right]d\Omega, \tag{19}$$

$$\tilde{\epsilon}_r''(\tilde{\nu}) = \left(\frac{2\tilde{\nu}}{\pi}\right) \mathcal{P} \int_0^\infty \left[\frac{\tilde{\epsilon}_r'(\Omega)}{\tilde{\nu}^2 - \Omega^2}\right] d\Omega, \tag{20}$$

where $\mathcal{P}$ denotes the Cauchy Principal Value. Equations (19) and (20) make it evident that $\tilde{\epsilon}_r'(\tilde{\nu})$ is an even function of $\tilde{\nu}$ and $\tilde{\epsilon}_r''(\tilde{\nu})$ is an odd function of $\tilde{\nu}$. The Kramers-Kronig relations also show that real and imaginary parts of $\tilde{\epsilon}_r(\tilde{\nu})$ are not independent of each other: if one is known, the other can be computed. Since the relationship between real and imaginary parts of $\tilde{\epsilon}_r(\tilde{\nu})$ is via integral transforms, computing $\tilde{\epsilon}_r'(\tilde{\nu})$, say, at a particular wavenumber $\tilde{\nu}$ from $\tilde{\epsilon}_r''(\tilde{\nu})$, requires knowledge of $\tilde{\epsilon}_r''(\tilde{\nu})$ over the entire range of wavenumbers from 0 to $\infty$. This makes it difficult to construct $\tilde{\epsilon}_r'(\tilde{\nu})$ from the experimentally measured imaginary part $\tilde{\epsilon}_r''(\tilde{\nu})$, if only the mid-IR range is measured, which is usually the case in experimental mid-IR absorption measurements. This, however, is not a problem for our reconstruction algorithm (to be presented in the following sections), since the algorithm is based on a set of analytical functions, given by the dielectric functions in Eqs (10) and (11), that exactly satisfy the Kramers-Kronig relations. We note that the complex refractive index $\tilde{n}(\tilde{\nu})$, derived via Eqs (14) and (15) from the complex permittivity $\tilde{\epsilon}_r(\tilde{\nu})$, also satisfies the analogous versions of the Kramers-Kronig relations, Eqs (19) and (20), exactly.

## 2.4 Inverse scattering problem

One way of measuring the chemical composition of a biological material is to homogenize it and then measure the absorbance caused by a thin film of the homogenized material. If the effect of fringes [42] (a form of scattering with well-known properties) can be removed, the result of this procedure directly yields the positions, heights, and widths of the absorption resonances of the functional groups contained in this material and can be used directly to identify the average chemical composition of the biological material under examination. However, unless the biological material of interest is inherently homogeneous, such as a sample of cytoplasm that can be easily spread into a thin film, this method is destructive. Consequently, it is not suitable for IR microspectroscopy of individual cells, where spatial resolution, i.e., the ability to detect the precise locations of different materials within a cell, is critical.

Since the size of individual biological cells is of the same order of magnitude as the incident IR radiation used in IR microspectroscopy, when an instrument records the missing intensity at the detector to determine the absorbance, the missing IR radiation is a consequence of both absorption and scattering of the radiation by the sample. Therefore, in contrast to spectroscopy performed on homogeneous thin films, the strong scattering contributions in this context disrupt the one-to-one correspondence between the measured absorbance and the absorption of radiation attributed to the sample's functional groups. Instead, the strong scattering phenomenon causes distorted baseline profiles, fringes, band distortions, shifts in peak positions, and changes in peak intensities. Consequently, a recorded absorbance spectrum does not reflect the pure absorbance that encodes the molecular composition of the cell. Instead, the measured IR absorbance spectrum is a superposition of the pure absorbance of the molecules and signals from scattering, as well as other optical phenomena, such as diffraction, that depend on the geometric cross-section of the sample and the wave nature of the IR radiation. Therefore, we need to remove the scattering contributions so that only the pure absorption due to the functional groups is recovered. This is accomplished most directly by solving an inverse scattering problem that automatically separates out the refractive and diffractive scattering effects from the pure absorption effects by determining (reconstructing) the complex permittivity $\tilde{\epsilon}_r(\tilde{\nu})$ or, equivalently, the complex index of refraction $\tilde{n}(\tilde{\nu})$, from

a given extinction efficiency $Q_{\text{ext}}^{(\text{given})}(\tilde{\nu})$, where $Q_{\text{ext}}^{(\text{given})}(\tilde{\nu})$, the input for the reconstruction of the permittivities (refractive indexes) may be experimentally measured or computed by a numerical simulation.

To illustrate the working principle and the power of our inverse scattering algorithm, it is not necessary that $Q_{\text{ext}}^{(\text{given})}(\tilde{\nu})$ is experimentally determined. Instead, we illustrate the capabilities of our inverse-scattering method following a two-step process. In the first step, we assume full knowledge of $\tilde{\epsilon}_r(\vec{r}, \tilde{\nu})$ $[\tilde{n}(\vec{r}, \tilde{\nu})]$ of our sample, and on the basis of this input compute the corresponding extinction efficiency, $Q_{\text{ext}}^{(\text{given})}(\tilde{\nu})$, numerically exactly. In the second step, starting from $Q_{\text{ext}}^{(\text{given})}(\tilde{\nu})$, we now run our inverse-scattering algorithm to (re-) discover (reconstruct) the $\tilde{\epsilon}_r(\vec{r}, \tilde{\nu})$ $[\tilde{n}(\vec{r}, \tilde{\nu})]$ we used to construct $Q_{\text{ext}}^{(\text{given})}(\tilde{\nu})$ in the first step. This two-step procedure exactly simulates an ideal IR microspectroscopy experiment including the analysis of the extinction efficiency, with the only difference that in the first step, instead of using an actual cell and an actual IR beam with a detector, we use a model cell and solve the electromagnetic Maxwell equations to obtain $Q_{\text{ext}}^{(\text{given})}(\tilde{\nu})$. We call the Maxwell-based computation of $Q_{\text{ext}}(\tilde{\nu})$ for any given $\tilde{\epsilon}_r(\tilde{\nu})$ the forward model. We have every confidence in the forward model, since, as mentioned in the introduction, biological cells are dielectric materials and it is well known that the Maxwell equations describe the scattering and absorption of dielectric materials exactly. For the purposes of this paper, we restrict ourselves to spherical model cells, homogeneously filled with a polymeric dielectric. In this case, we can use a publicly available Mie scattering code, PyMieScatt (in Python), to numerically solve the exact forward model. What remains to be explained now are the details of our algorithm for performing the second (analysis) step, which is presented in the following section.

## 2.5 Solution algorithm for the inverse scattering problem

For multi-electron systems, exhibiting multiple resonances, the final form of the complex permittivity can be written as a sum over all absorption bands. Labeling the absorption bands with the index $m$ for $m = 1, \dots, M$, where $M$ is the total number of absorption bands in the mid IR range, we extend the real and imaginary parts of $\tilde{\epsilon}_r(\tilde{\nu})$, given in Eqs (10) and (11), respectively, to the case with $M$ absorption bands according to

$$\tilde{\epsilon}_r{}'(\tilde{\nu}) = \epsilon_\infty + \sum_{m=1}^{M} \left\{ \frac{\tilde{\nu}_p^{(m)^2}(\tilde{\nu}_0^{(m)^2} - \tilde{\nu}^2)}{(\tilde{\nu}_0^{(m)^2} - \tilde{\nu}^2)^2 + (\tilde{\nu}\gamma^{(m)})^2} \right\}, \tag{21}$$

$$\tilde{\epsilon}_r{}''(\tilde{\nu}) = \sum_{m=1}^{M} \left\{ \frac{\tilde{\nu}\gamma^{(m)}\tilde{\nu}_p^{(m)^2}}{(\tilde{\nu}_0^{(m)^2} - \tilde{\nu}^2)^2 + (\tilde{\nu}\gamma^{(m)})^2} \right\}, \tag{22}$$

where the parameters, $\tilde{\nu}_0^{(m)}, \tilde{\nu}_p^{(m)}, \gamma^{(m)}$, represent the resonant wavenumber, peak height, and full width at half maximum, respectively, of IR absorption band number $m$. As discussed in the previous section, given the complex permittivity of any one of our six test substances, we first generate its corresponding extinction efficiency, $Q_{\text{ext}}(\tilde{\nu})$, in the mid IR region, using the forward model. We take this extinction efficiency to be the given extinction efficiency, $Q_{\text{ext}}^{(\text{given})}(\tilde{\nu})$. The reconstruction algorithm then works in the following way:

Find the optimal set of parameters, $\tilde{\nu}_0^{(m)}, \tilde{\nu}_p^{(m)}$, and $\gamma^{(m)}$, $m = 1, \dots, M$, that minimize the residual sum of squares, $S$, between the given extinction efficiency and the fitted extinction

efficiency,

$$S\left[\left\{\tilde{\nu}_0^{(m)}, \tilde{\nu}_p^{(m)}, \gamma^{(m)}\right\}\right] = \sum_{l=1}^{L} R^2\left[\left\{\tilde{\nu}_0^{(m)}, \tilde{\nu}_p^{(m)}, \gamma^{(m)}\right\}, \tilde{\nu}_l\right], \qquad (23)$$

where $\tilde{\nu}_l$ are the wavenumber mesh points provided in the literature at which the refractive indexes were measured (typically, $L \sim 600$), and

$$R\left[\left\{\tilde{\nu}_0^{(m)}, \tilde{\nu}_p^{(m)}, \gamma^{(m)}\right\}, \tilde{\nu}\right] = Q_{ext}^{(model)}\left[\left\{\tilde{\nu}_0^{(m)}, \tilde{\nu}_p^{(m)}, \gamma^{(m)}\right\}, \tilde{\nu}\right] - Q_{ext}^{(given)}(\tilde{\nu}) \qquad (24)$$

are the residuals and $Q_{ext}^{(model)}\left[\left\{\tilde{\nu}_0^{(m)}, \tilde{\nu}_p^{(m)}, \gamma^{(m)}\right\}, \tilde{\nu}\right]$ is the extinction efficiency computed by the forward model at wavenumber $\tilde{\nu}$ for the parameters $\tilde{\nu}_0^{(m)}, \tilde{\nu}_p^{(m)}, \gamma^{(m)}, m = 1, 2, \ldots, M$. The notation $\{\tilde{\nu}_0^{(m)}, \tilde{\nu}_p^{(m)}, \gamma^{(m)}\}$ denotes the entire set of variational parameters, $\tilde{\nu}_0^{(m)}, \tilde{\nu}_p^{(m)}, \gamma^{(m)}$, for $m = 1, 2, \ldots, M$.

As a result of the above algorithm, $Q_{ext}^{(model)}(\tilde{\nu})$ reproduces $Q_{ext}^{(given)}(\tilde{\nu})$ to a good approximation, which then, via the optimal set of parameters $\{\tilde{\nu}_0^{(m)}, \tilde{\nu}_p^{(m)}, \gamma^{(m)}\}$, results in the reconstructed complex permittivity $\tilde{\epsilon}_{r,\text{fit}}(\tilde{\nu})$ [$(\tilde{n}_{\text{fit}}(\tilde{\nu})$, respectively], which then, because of the quality of the fit of $Q_{ext}^{(model)}(\tilde{\nu})$ to $Q_{ext}^{(given)}(\tilde{\nu})$, is also a good approximation to the real and imaginary components of the experimental permittivity $\tilde{\epsilon}_r(\tilde{\nu})$ [$\tilde{n}(\tilde{\nu})$, respectively] of the material under investigation. The results of this algorithm and the quality of the fits are presented in Section 3.

## 2.6 Materials and IR measurements

Absorption bands in a biological sample may represent functional groups associated with proteins, lipids, and carbohydrates that occur primarily in the ranges $900\,\text{cm}^{-1}$ – $1200\,\text{cm}^{-1}$ ($11.111\,\mu\text{m}$–$8.333\,\mu\text{m}$), $1600\,\text{cm}^{-1}$–$1700\,\text{cm}^{-1}$ ($6.250\,\mu\text{m}$–$5.882\,\mu\text{m}$), and around $1740\,\text{cm}^{-1}$ ($5.747\mu\text{m}$) [43]. While the relative intensities of the spectral bands depend on the concentration of cellular molecules, any shift in the locations of the bands may represent changes in the molecular composition of the cells, indicating a potential malignancy, and is therefore an important clinical marker. Thus, to develop a method that can account for a multitude of functional behaviors and resonances in the sample analyzed over the IR spectrum, and still reconstruct the molecules accurately, the algorithm must be tested and validated with a range of different polymers. Consequently, for our analysis, we decided to investigate a set of six different organic polymers that serve as test substances for the reconstruction method: polymethyl methacrylate (PMMA), polycarbonate (PC), polydimethylsiloxane (PDMS), polyetherimide (PEI), polyethylene terephthalate (PET), and polystyrene (PS) with molecular repeating units: $[C_5H_8O_2]_n$, $[C_{16}H_{14}O_3]_n$, $[C_2H_6OSi]_n$, $[C_{37}H_{24}O_6N_2]_n$, $[C_{10}H_8O_4]_n$, and $[C_8H_8]_n$, respectively. Although they are members of the same polymer class, composed of carbon chain backbones, the substances have their unique properties emerging from differences in their functional groups, which result in distinct dielectric responses. The extinction spectrum of PMMA shows peaks between $2800\,\text{cm}^{-1}$–$3000\,\text{cm}^{-1}$ ($3.571\,\mu\text{m}$–$3.333\,\mu\text{m}$), assigned to methylene and methyl groups of $CH_2$ and $CH_3$, at $1730\,\text{cm}^{-1}$ ($5.780\,\mu\text{m}$) for the C=O band, and around $1000\,\text{cm}^{-1}$ – $1300\,\text{cm}^{-1}$ ($10.000\,\mu\text{m}$–$7.692\,\mu\text{m}$), corresponding to the C–O band [44]. The infrared-active vibrational modes of PC include C–H and C=O stretching at $2970\,\text{cm}^{-1}$ ($3.367\,\mu\text{m}$) and $1772\,\text{cm}^{-1}$ ($5.643\,\mu\text{m}$), respectively, C–C ring stretching at $1500\,\text{cm}^{-1}$ ($6.667\,\mu\text{m}$), C–C–C bending at $1080\,\text{cm}^{-1}$ ($9.259\,\mu\text{m}$), O–C–O stretching at $1015\,\text{cm}^{-1}$ ($9.852\,\mu\text{m}$), and out-of-plane C–H deformation at $830\,\text{cm}^{-1}$ ($12.048\,\mu\text{m}$) [45].

The absorption peaks of PET are attributed to aromatic and aliphatic C–H bond stretching between $2800 \, \text{cm}^{-1}$–$3100 \, \text{cm}^{-1}$ ($3.571 \, \mu\text{m}$–$3.226 \, \mu\text{m}$), ester carbonyl bond stretching at $1720 \, \text{cm}^{-1}$ ($5.814 \, \mu\text{m}$), ester group stretching at $1300 \, \text{cm}^{-1}$ ($7.692 \, \mu\text{m}$), and a methylene group at $1100 \, \text{cm}^{-1}$ ($9.091 \, \mu\text{m}$) [46]. PDMS typically demonstrates IR peaks in the range of $789 \, \text{cm}^{-1}$–$800 \, \text{cm}^{-1}$ ($12.674 \, \mu\text{m}$–$12.500 \, \mu\text{m}$), corresponding to $CH_3$ rocking and Si–C stretching in Si–$CH_3$, $1020 \, \text{cm}^{-1}$–$1094 \, \text{cm}^{-1}$ ($9.804 \, \mu\text{m}$–$9.141 \, \mu\text{m}$) due to the Si–O–Si stretching, $1260 \, \text{cm}^{-1}$–$1259 \, \text{cm}^{-1}$ ($7.937 \, \mu\text{m}$–$7.943 \, \mu\text{m}$) corresponding to the $CH_3$ deformation in Si–$CH_3$, and $2850 \, \text{cm}^{-1}$–$2960 \, \text{cm}^{-1}$ ($3.509 \mu\text{m}$–$3.378 \mu\text{m}$) for the symmetric and asymmetric C–H stretching [47]. The extinction spectrum of PEI exhibits the characteristic absorption of the imide carbonyl asymmetric stretch at $1720 \, \text{cm}^{-1}$ ($5.814 \, \mu\text{m}$) and symmetric stretch at $1780 \, \text{cm}^{-1}$ ($5.618 \, \mu\text{m}$), for C–N stretching and bending at $1355 \, \text{cm}^{-1}$ ($7.380 \, \mu\text{m}$) and $743 \, \text{cm}^{-1}$ ($13.459 \, \mu\text{m}$), respectively, and for aromatic ether C–O–C at $1234 \, \text{cm}^{-1}$ ($8.104 \, \mu\text{m}$) [48]. Lastly, PS generates absorption bands in the region $3025 \, \text{cm}^{-1}$ – $3081 \, \text{cm}^{-1}$ ($3.306 \, \mu$–$3.246 \, \mu\text{m}$) due to the aromatic C–H stretch, $2923 \, \text{cm}^{-1}$–$2850 \, \text{cm}^{-1}$ ($3.421 \, \mu\text{m}$–$3.509 \, \mu\text{m}$) for the $CH_2$ asymmetric and symmetric stretches (methylenes), $1452 \, \text{cm}^{-1}$ – $1600 \, \text{cm}^{-1}$ ($6.887 \, \mu\text{m}$–$6.250 \, \mu\text{m}$) for the aromatic C=C stretching modes, indicating the presence of benzene rings, and $756 \, \text{cm}^{-1}$–$698 \, \text{cm}^{-1}$ ($13.227 \, \mu\text{m}$–$14.327 \, \mu\text{m}$) for the vibration of C–H out-of-plane bending, verifying the existence of a single substituent in the benzene ring [49].

The refractive indexes of the materials were obtained from the literature [50], where the data was experimentally measured from slabs of the respective materials with thicknesses characterized using ellipsometry and FT-IR microscopy. In our reconstruction simulations, as described in the previous section, this refractive-index data was then used to compute the respective permittivities and the extinction efficiency, $Q_{ext}^{(given)}$, for dielectric spheres, which encodes information not only about the structure of the material but also about the size of the sample. A constant sphere radius of five microns was used for all six materials.

## 3 Results and discussion

In Section 2, we outlined our methods by providing a detailed description of our inverse-scattering reconstruction algorithm. The purpose of this section is to illustrate its capabilities. We start in Section 3.1 by documenting the suitability and efficiency of the dielectric functions, defined in Section 2.2, that underlie the forward model, defined in Section 2.4. In Section 3.2, we present the reconstructions of the complex permittivities of the six test polymers. Excellent agreement is obtained between the experimental permittivities of the six test polymers and their reconstructions. While from the outset (see Section 2.2), we settled on the exact dielectric functions as our preferred basis functions, we add in Section 3.3 a few more arguments and numerical results, indicating, from a practical point of view, that the dielectric functions are indeed superior to the Lorentzian functions as the basis set for these reconstruction optimizations.

### 3.1 Quality of the basis

Since the goal of our reconstruction algorithm is the extraction of the underlying complex dielectric permittivity from a given extinction efficiency $Q_{ext}^{(given)}(\tilde{\nu})$, it is necessary that our basis of dielectric functions has the power to closely represent the permittivity that gives rise to $Q_{ext}^{(given)}(\tilde{\nu})$ via the forward model. Since any additional basis function included in the algorithm has the potential to increase the accuracy of the reconstruction, it is desirable to include as many basis functions as possible in the reconstruction algorithm. However, we have to be economical with the number of basis functions included, since any additional basis function increases the complexity of the optimization and thus the execution time of the algorithm.

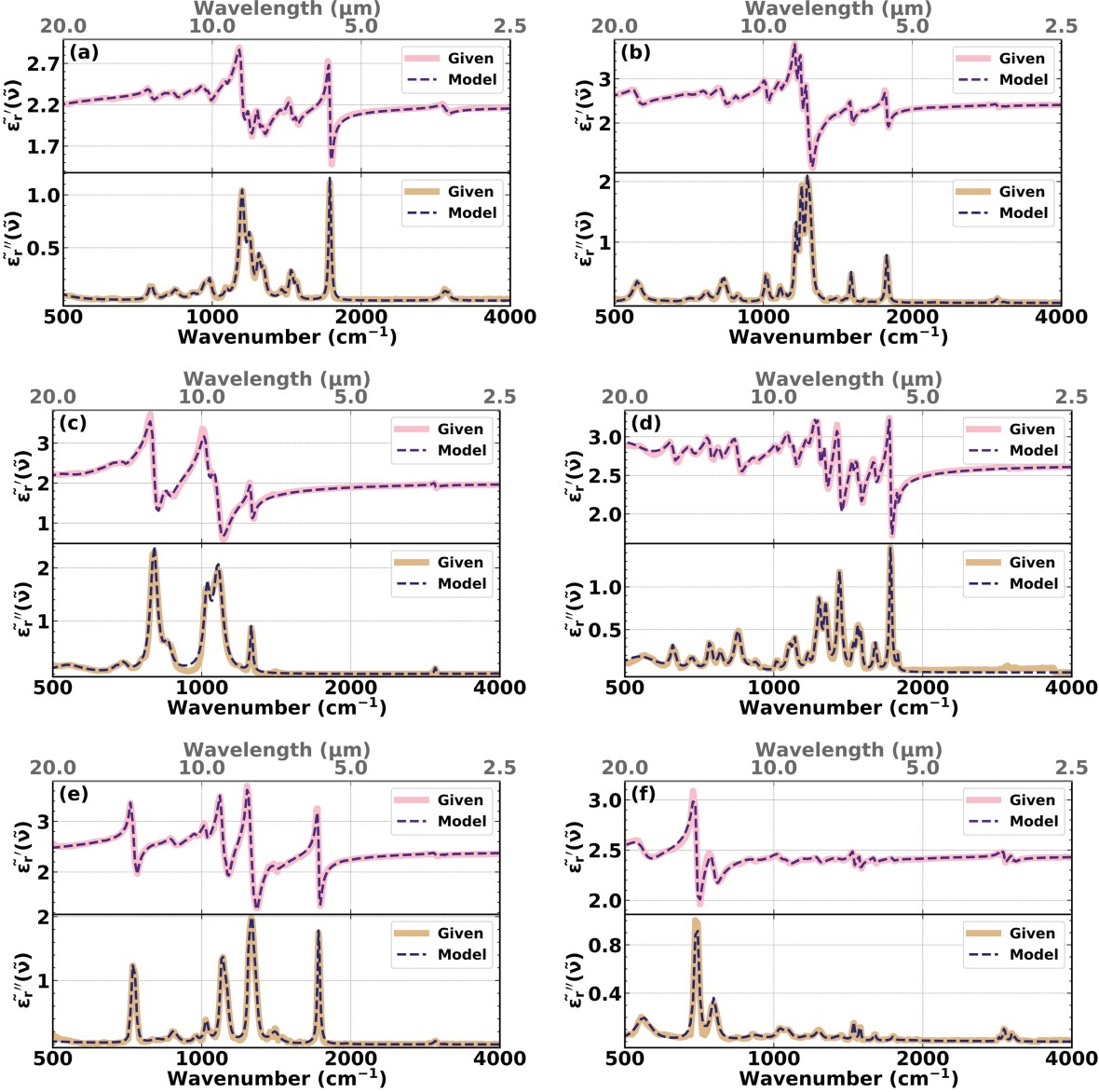

**Fig 1. Real and imaginary parts of the complex permittivities, $\tilde{\varepsilon}_r(\tilde{\nu})$, for the six test polymers: (a) PMMA, (b) PC, (c) PDMS, (d) PEI, (e) PET, and (f) PS.** The top (bottom) half of each of the six panels shows the experimental real (imaginary) parts of the respective polymer (solid pink and solid burlywood, respectively) together with their fits based on a superposition of optimized dielectric functions (dashed indigo and dashed blue, respectively).

Taking these considerations into account and running numerous explorative reconstructions, we found that a minimum of 17 basis functions are necessary to fit the experimental complex permittivities of our six test substances. Therefore, to demonstrate the quality of our basis set, we fitted the experimental complex permittivities of our six test substances (PMMA, PC, PDMS, PEI, PET, and PS) with 17 dielectric basis functions. The result is shown in Fig 1. It shows a comparison between the complex experimental permittivities (solid lines) of the respective materials and their fits (dashed lines) for the real (top) and imaginary (bottom) parts. It can be seen that already with a minimum of 17 basis functions excellent fits of the permittivities can be obtained. Of course, as discussed above, the accuracy only increases with

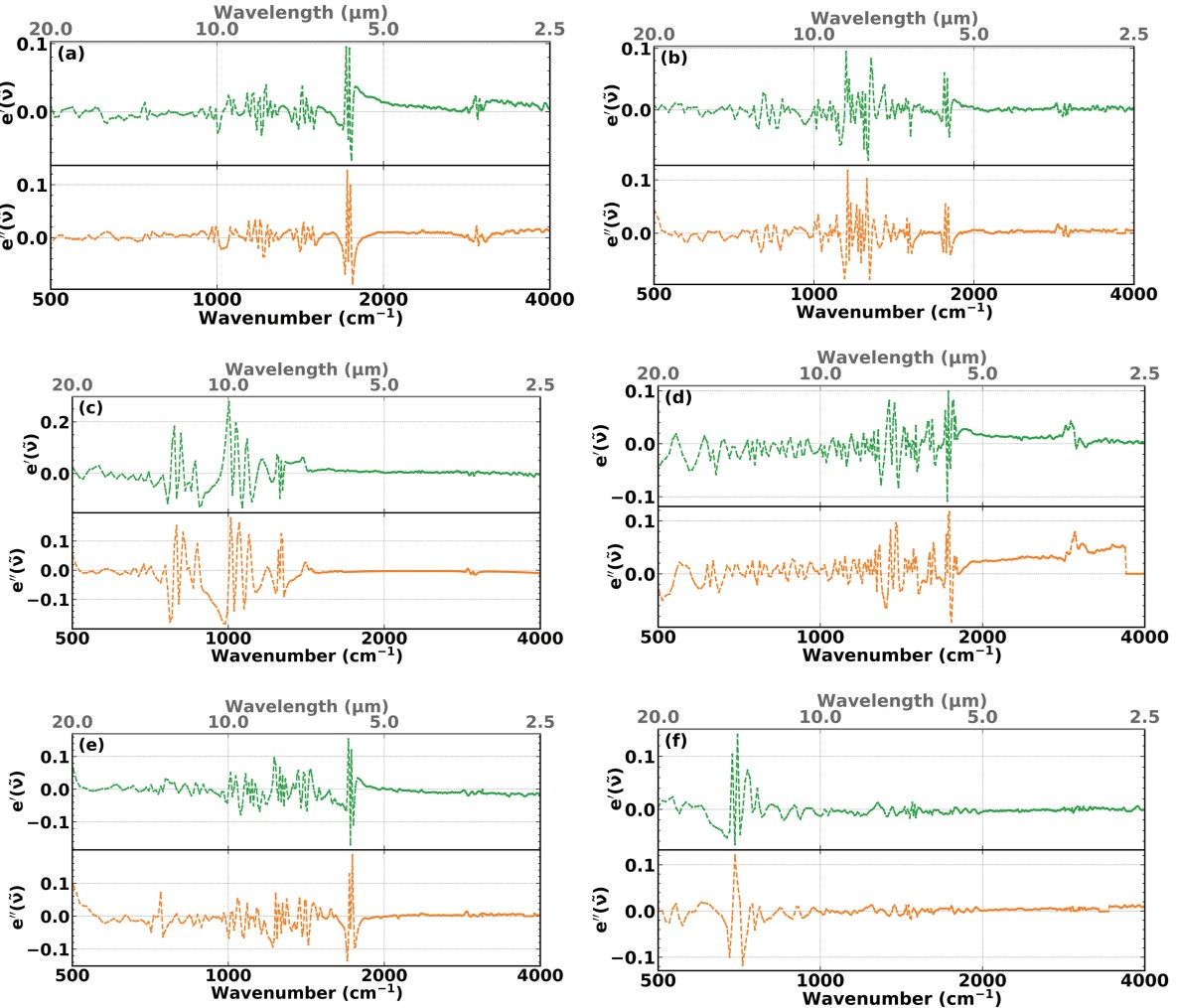

**Fig 2. Residual errors,** $e'(\tilde{\nu})$ **(green) and** $e''(\tilde{\nu})$ **(orange), in the fits of the six test polymers to their respective experimental permittivities.** (**a**) PMMA, (**b**) PC, (**c**) PDMS, (**d**) PEI, (**e**) PET, and (**f**) PS.

an increase of the number of basis functions, which is one of the reasons why we increased the number of basis functions to 25 in the following Section 3.2.

To quantitatively assess the quality of the 17-basis-functions fits, we define the residual errors according to

$$e'(\tilde{\nu}) = \tilde{\epsilon}'_{r,\text{exp}}(\tilde{\nu}) - \tilde{\epsilon}'_{r,\text{fit}}(\tilde{\nu}), \tag{25}$$

$$e''(\tilde{\nu}) = \tilde{\epsilon}''_{r,\text{exp}}(\tilde{\nu}) - \tilde{\epsilon}''_{r,\text{fit}}(\tilde{\nu}), \tag{26}$$

where $\tilde{\epsilon}'_{r,\text{exp}}(\tilde{\nu})$ and $\tilde{\epsilon}''_{r,\text{exp}}(\tilde{\nu})$ are the experimental permittivites and $\tilde{\epsilon}'_{r,\text{fit}}(\tilde{\nu})$ and $\tilde{\epsilon}''_{r,\text{fit}}(\tilde{\nu})$ are the corresponding fitted permittivities. The residual-error plots for the curve fits of the six test polymers are shown in Fig 2. Overall the fit errors are very small. In addition, Fig 2 shows that the residuals appear to fluctuate randomly around zero, indicating that the fit is good and that the model describes the data well.

## 3.2 Reconstruction of complex permittivities from extinction spectra

Having determined that the dielectric functions are an effective basis set for the non-linear reconstruction algorithm, we now discuss the process of choosing a suitable set of initial conditions for extracting the optimal parameters, $\tilde{\nu}_0^{(m)}$, $\tilde{\nu}_p^{(m)}$, and $\gamma^{(m)}$, associated with the functional groups of the material. Most organic polymers, particularly the ones that we investigate in this paper, demonstrate roughly between 15 to 20 absorption bands of substantial intensities in the mid-IR frequency spectrum. While starting with a large number of dielectric functions can improve the reconstruction accuracy, it also increases the size and complexity of the optimization problem, thereby reducing efficiency. On the other hand, a less than sufficient number of dielectric functions in the basis set can lead to the loss of chemical information by missing significant absorption bands. To ensure that all absorption bands are captured during reconstruction, we settled on a basis set of 25 dielectric functions. Since each dielectric function comprises three parameters, this translates to solving an inverse optimization problem in a parameter space of 75 dimensions.

Once the size of the basis set is chosen, we need to select initial conditions for $\tilde{\nu}_0^{(m)}$, $\tilde{\nu}_p^{(m)}$, and $\gamma^{(m)}$ to start our reconstruction algorithm. How to best select the initial conditions depends on the degree of prior knowledge we have about the material to be analyzed. In the following two sections, we consider two cases. In the first case, called "Case I," discussed in Section 3.2.1, we assume that we have no prior knowledge of the substance at all. In the second case, called "Case II," discussed in Section 3.2.2, we assume that the substance to be analyzed is a polymer, and since we have a good idea of how a generic mid-IR spectrum of an organic polymer looks like, we choose our initial conditions accordingly.

**3.2.1 Case I: reconstruction with no prior knowledge of the material's absorption spectrum.** We place 25 dielectric functions, equi-spaced in $\tilde{\nu}$, with $\tilde{\nu}_p^{(m)} = 20\,\text{cm}^{-1}$ and $\gamma_p^{(m)} = 20\,\text{cm}^{-1}$, $m = 1, \dots, M$, in the frequency interval of $500\,\text{cm}^{-1}$ to $4000\,\text{cm}^{-1}$ ($20.0\,\mu\text{m}$ to $2.50\,\mu\text{m}$), the same for all six test substances. This guarantees that the starting conditions for the control parameters are unbiased. Positioning the peaks equidistantly ensures that a dielectric basis function is always near an actual absorption band in the mid-IR spectrum, thus allowing the reconstruction algorithm to fit basis functions to the nearest absorption peak with minor adjustments to the control parameters. This feature significantly reduces the chance of getting trapped in local minima and inflection points, contributing to an overall improvement of the fit quality.

Fig 3 shows the fit of the real (dashed blue) and imaginary (dashed purple) parts of the permittivity and the extinction efficiency (dashed red) to the experimental data (solid pale green, solid khaki, and solid light blue, respectively) for the different materials (PMMA, PC, PDMS, PEI, PET, and PS, respectively). One can see that there is a significant overlap between the fitted data (model) and the experiment (given), indicating that the reconstruction algorithm was effective in extracting the absorption bands associated with the major functional groups in the model cells. This is all the more surprising considering that the same, unbiased, initial conditions were used for all six test substances.

In Fig 3, and also in Fig 4 (see the following Section 3.2.2), in addition to showing the complex effective permittivities $\tilde{\varepsilon}_r(\tilde{\nu})$, we also show the extinction efficiencies $Q_{\text{ext}}(\tilde{\nu})$, which allows direct comparisons between the features (such a peaks) in $\tilde{\varepsilon}_r(\tilde{\nu})$ and $Q_{\text{ext}}(\tilde{\nu})$. While in fringe-corrected thin-film infrared spectroscopy [42] there is a nearly one-to-one relationship between peaks in the imaginary part of the permittivity and peaks in the extinction spectrum, because of the scattering contributions of finite-size samples, such as cells or spheres, scattering contributions distort this nearly one-to-one correspondence. However, as shown in Fig 3 and in Fig 4 (see Section 3.2.2 below), vestiges of this relationship remain. This correlation,

however, as illustrated by Figs 3 and 4, is too vague to form the basis of an accurate reconstruction method. The full inverse-scattering, numerical method, as outlined in our paper, is required.

**3.2.2 Case II: reconstruction with some prior knowledge of the material's absorption spectrum.** The quality of the reconstruction can be enhanced by making use of any relevant information available on the investigated material. For instance, in the case of organic substances, it is known that most mid-IR absorption bands associated with the major functional groups tend to occur in the range between $500\,\mathrm{cm}^{-1}$ and $4000\,\mathrm{cm}^{-1}$ ($20.00\,\mu$m and $2.500\,\mu$m, respectively). This can roughly be divided into four intervals: **(a)** $500\,\mathrm{cm}^{-1}$–$2200\,\mathrm{cm}^{-1}$ ($20.00\,\mu$m–$4.455\,\mu$m), where significant absorption occurs, **(b)** $2200\,\mathrm{cm}^{-1}$–$2800\,\mathrm{cm}^{-1}$ ($4.455\,\mu$m–$3.571\,\mu$m), where little to no absorption occurs (silent interval I), **(c)** $2800\,\mathrm{cm}^{-1}$–$3500\,\mathrm{cm}^{-1}$ ($3.571\,\mu$m–$2.857\,\mu$m), where absorption is again detected, and **(d)** $3500\,\mathrm{cm}^{-1}$–$4000\,\mathrm{cm}^{-1}$ ($2.857\,\mu$m–$2.500\,\mu$m), where, again, little to no absorption takes place (silent interval II). With this knowledge in hand, we can now narrow down the space of initial parameter settings by focusing on the regions where we expect absorption bands to be present (intervals **a** and **c**) and allocate relatively fewer of our resources in the two silent intervals I and II, where the chances of encountering absorption peaks are small (intervals **b** and **d**).

We test this procedure by placing 25 dielectric functions with $\tilde{\nu}_p^{(m)} = 20\,\mathrm{cm}^{-1}$ and $\gamma_p^{(m)} = 20\,\mathrm{cm}^{-1}$, $m = 1, \ldots, M$, in the frequency interval of $500\,\mathrm{cm}^{-1}$ to $4000\,\mathrm{cm}^{-1}$ ($20.0\,\mu$m to $2.50\,\mu$m), the same for all six test substances. Moreover, we divide the set of dielectric functions into two groups, the first consisting of 20 dielectric functions, evenly distributed over the interval **a**, and the second consisting of the remaining five dielectric functions, evenly distributed over the interval **c**.

The resulting reconstructions of the real (dashed blue) and imaginary (dashed purple) parts of the permittivities compared with the experimental permittivities (solid pale green and solid khaki, respectively), and the fits of the corresponding extinction efficiencies $Q_{\mathrm{ext}}^{(\mathrm{model})}$ (dashed red) to the experimental extinction efficiencies, $Q_{\mathrm{ext}}^{(\mathrm{given})}$, (solid light blue) of PMMA, PC, PDMS, PEI, PET, and PS, respectively, are shown in Fig 4. We find that the agreement between the fitted and the experimental curves is excellent and the difference between the results is visually almost indistinguishable, especially when it comes to reconstructing the major functional groups. It can also be observed that the reconstructions obtained in Case II are better at capturing the structural and molecular properties of the model cells than those obtained in Case I, which is an added benefit of using prior knowledge to constrain the space of initial conditions. For PMMA spheres reconstructions are also reported in the literature (see, e.g., [16,34,51]). Compared with the literature reconstructions the quality of our reconstructions, as shown in Figs 3 and 4, is superior. In addition, our reconstructions are more comprehensive since we tested our method for six different substances while the literature focuses on reconstructions for PMMA spheres. Moreover, our method, since it is an ab initio, direct, inverse scattering method, does not require advanced spectroscopic expertise, such as choosing a reference spectrum, as is required for other reconstruction methods (see, e.g., [18]).

## 3.3 Dielectric functions versus anti-symmetrized Lorentzians

We previously developed a reconstruction algorithm [31] that expresses the complex refractive index as a sum of anti-symmetrized Lorentzians and optimizes the parameters of the anti-symmetrized Lorentzians that best fit the space- and frequency-dependent refractive indexes associated with the absorbance spectrum of the two materials used, i.e., PMMA and PS. It was found that the Lorentzian basis-set for the reconstruction algorithm worked well to extract

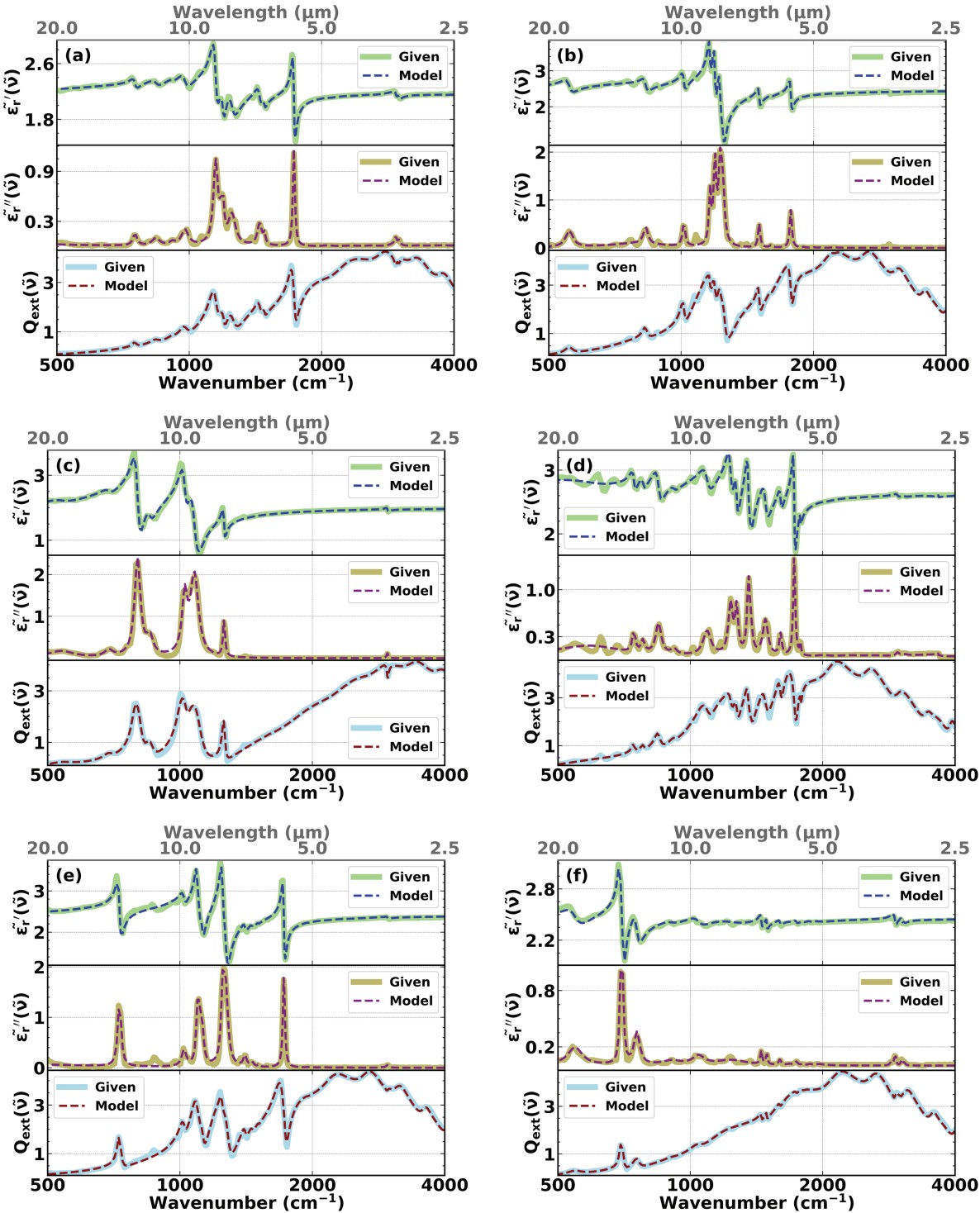

**Fig 3. Reconstructions of the real (top of each of the six frames) and imaginary (middle of each of the six frames) parts of the experimental permittivities, $\tilde{\epsilon}_r{}'(\tilde{\nu})$ and $\tilde{\epsilon}_r{}''(\tilde{\nu})$, respectively, and the fit of $Q_{ext}^{(given)}(\tilde{\nu})$ (bottom of each of the six frames) of (a) PMMA, (b) PC, (c) PDMS, (d) PEI, (e) PET, and (f) PS for the choice of initial conditions that characterize Case I.**

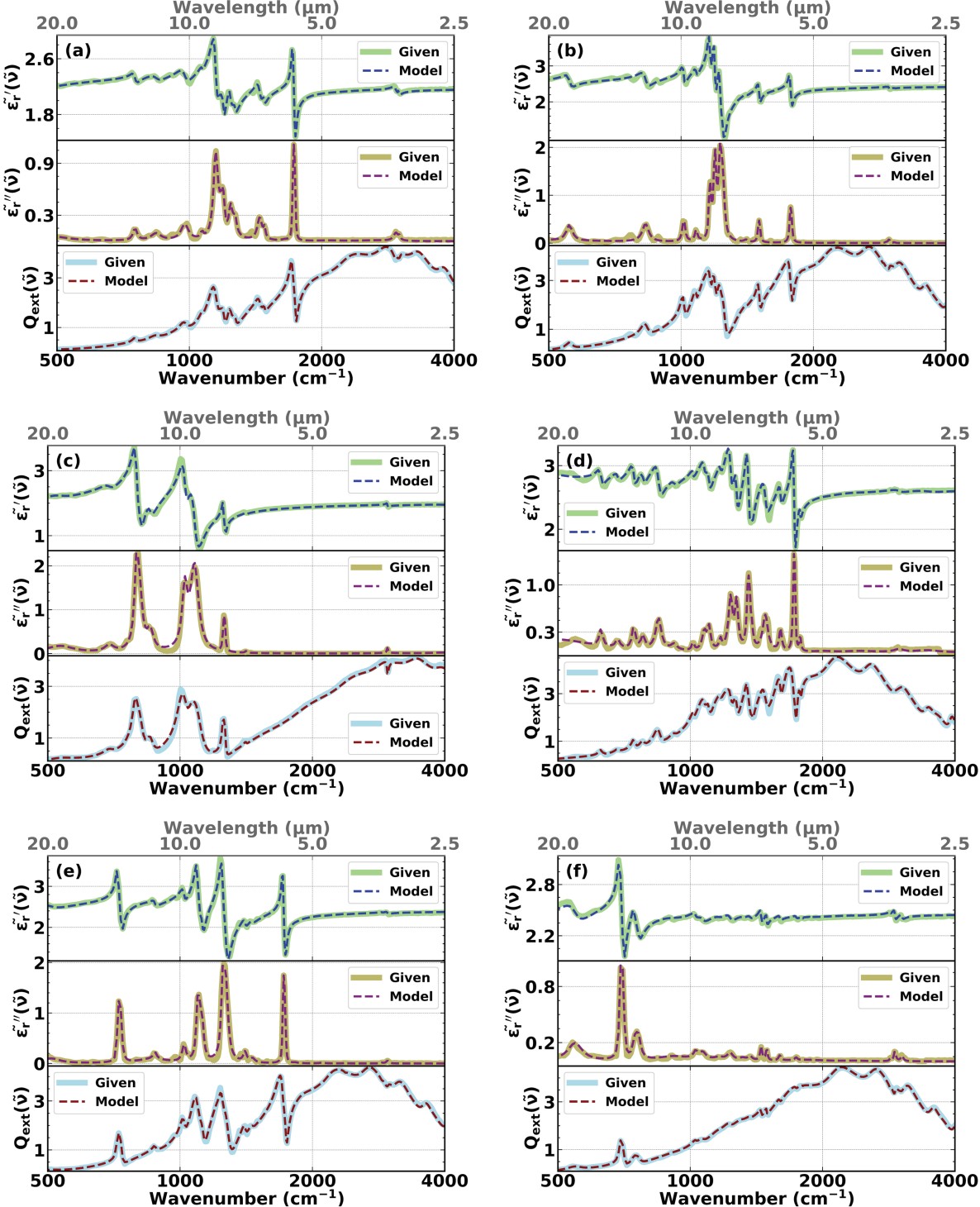

**Fig 4. Reconstructions of the real (top of each of the six frames) and imaginary (middle of each of the six frames) parts of the experimental permittivities, $\tilde{\epsilon}_r{}'(\tilde{\nu})$ and $\tilde{\epsilon}_r{}''(\tilde{\nu})$, respectively, and the fit of $Q_{ext}^{(given)}(\tilde{\nu})$ (bottom of each of the six frames) of (a) PMMA, (b) PC, (c) PDMS, (d) PEI, (e) PET, and (f) PS for the choice of initial conditions that characterize Case II.**

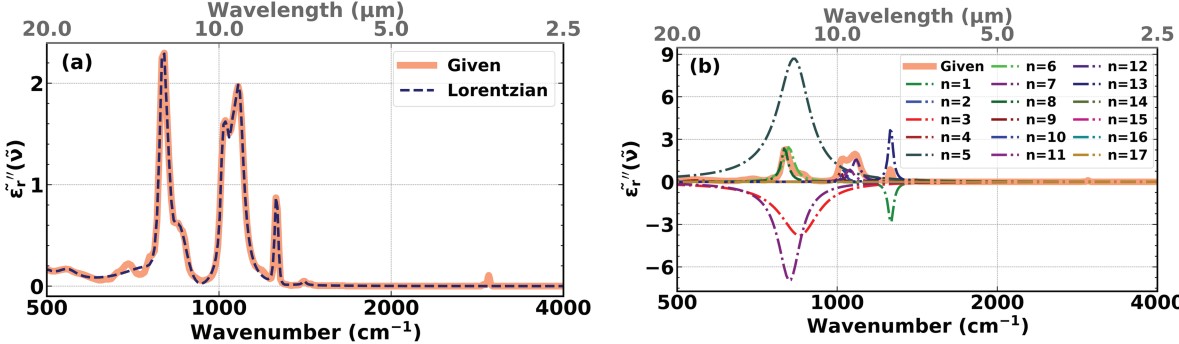

**Fig 5. (a)** Imaginary part, $\tilde{\epsilon_r}''(\tilde{\nu})$, of the experimental permittivity of PDMS (solid orange) compared with its fitted version (dashed dark blue), constructed from a sum of 17 optimized, anti-symmetrized Lorentzian functions. **(b)** The functional form of each of the 17 individual Lorentzian functions, labelled $n = 1$ to $n = 17$, which make up the complete fitted function (dashed dark blue) in **(a)**. Some of the 17 individual contributions are negative.

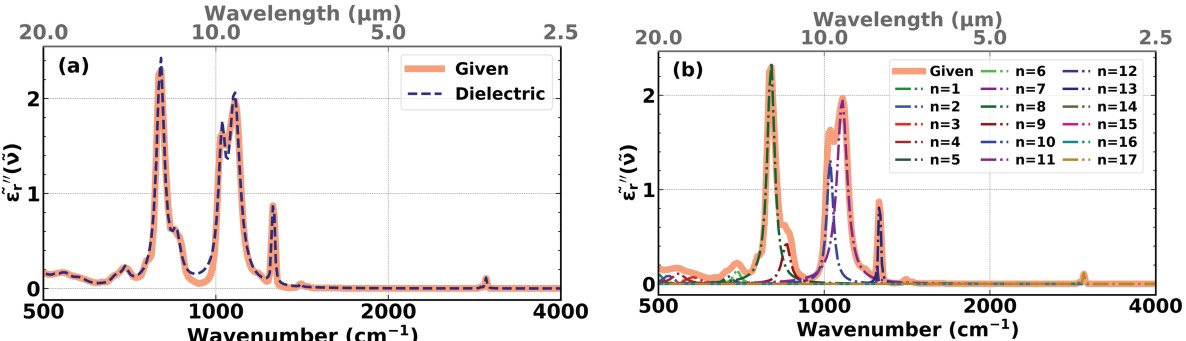

**Fig 6. (a)** Imaginary part, $\tilde{\epsilon_r}''(\tilde{\nu})$, of the experimental permittivity of PDMS (solid orange) compared with its fitted version (dashed dark blue), constructed from a sum of 17 optimized dielectric functions. **(b)** The functional form of each of the 17 individual dielectric functions, labelled $n = 1$ to $n = 17$, which make up the complete fitted function (dashed dark blue) in **(a)**. None of the 17 individual contributions is negative.

chemical information and was robust concerning perturbations to initial conditions and noise. Despite producing good reconstructions of the functional groups in PMMA and PS, we believe that the Lorenztians are not the best basis set to be used for the reconstruction of biological cells since they are not the natural functions representing the dielectric response of functional groups. This is made clear, e.g., from the material reconstruction of PDMS, where the optimized parameters assume negative values (Fig 5), especially in regions of overlapping resonances, where the electromagnetic wave experiences constructive and destructive inference. This may cause numerical instability induced by the different Lorentzians exhibiting positive and negative amplitudes to result in optimal fits. Even though the reconstruction of the molecular composition of the substances is hardly compromised using Lorentzians, they fail to represent the true resonances of the system, and instead simply act as mathematical tools, i.e., fit functions for the dielectric response. Thus, using Lorentzians only allows reproducing the complex refractive index but does not illuminate the underlying physics governing the absorption bands that reflect the material properties.

In contrast to reconstructions with Lorentzian basis functions that always resulted in negative parameter values, we found that working with dielectric basis functions, we were always

able to find excellent reconstructions with only positive parameters. An example is shown in Fig 6 for the same material as in Fig 5. We observe the quality of the reconstruction (Fig 6a) and the absence of any negative parameter values (Fig 6b). Looking more closely and comparing the reconstructions in Figs 5 and 6, we notice that working with the Lorentzian basis (Fig 5), the reconstruction misses the peak at around $750\,\mathrm{cm}^{-1}$, while this peak is perfectly captured using the dielectric basis (Fig 6). Conversely, the dip at around $950\,\mathrm{cm}^{-1}$ is better captured using the Lorentzian basis. However, missing a peak is much more serious than not precisely capturing a broad valley. Therefore, the reconstruction in Fig 6 is superior to the reconstruction in Fig 5, which is based on the Lorentzian functions.

Thus, for the following reasons, dielectric functions are the more appropriate basis to use for the reconstruction of structural and chemical properties of these model cells:

1. They are the natural functions for the electromagnetic response of matter reflected in the non-negative Lorentz-model parameters as confirmed in the reconstructions (Fig 6), which makes computations numerically more stable. In addition,
2. they automatically satisfy the Kramers-Kronig relations and therefore,
3. offer analytical functions for the real and imaginary parts of the complex permittivity,
4. which are Hilbert Transforms of each other, and thus do not require the explicit (numerical) use of Kramers-Kronig transformations.

In Figs 5 and 6 we show the results for 17 basis functions, since we found in Section 3.1 that 17 basis functions are sufficient for good accuracy, and showing the results for 25 basis functions would make the figures too crowded. Nevertheless, we conducted additional simulations with 18, …, 25 basis functions and obtained results similar to the ones shown in Figs 5 and 6.

## 4 Summary and conclusions

The central task in IR microspectroscopy is to extract (reconstruct) the complex permittivity of a microscopic inorganic or organic sample, for instance a biological cell, from a measured absorbance or extinction spectrum. Because of the microscopic size of the sample, usually of the order of the IR wavelength itself, scattering effects are large, distorting absorption signals. Therefore, traditional methods of reconstruction work akin to a two-step approach, where the reconstructed signal is scatter corrected using advanced correction algorithms that may be based on machine learning. In this paper, we present a different approach where scattering effects are integrated into the reconstruction algorithm from the outset, as formulated in our previous work [31], rather than being corrected in a subsequent step. Our proposed method can therefore be referred to as a direct approach to reconstruction since both scattering and absorption effects are taken into account simultaneously. One advantage of this direct method is that, in principle, it is mathematically exact, i.e., the global optimum of the set of dielectric functions is contained in the solution space and reproduces the given absorbance- or extinction spectrum exactly. Conversely, the challenge with a direct approach lies in finding optimal solutions in high-dimensional variational spaces, which, in the examples discussed in this paper, can have up to 75 dimensions. Nonlinear optimization in such high-dimensional spaces is well known to be risky, as the optimization algorithm may easily get trapped in local minima that are far from the optimal solution. It therefore came as a surprise to us that for all the six test substances investigated in this paper, the nonlinear optimization in the reconstruction finds a set of dielectric functions whose superposition closely reproduces the known, experimental permittivities. This demonstrates the strength of our reconstruction method,

which requires no scattering correction and yields the reconstructed dielectric functions directly from the optimization algorithm. Another advantage of our method is its independence, in principle, from the specific basis set used for reconstruction, provided the basis set is complete and Kramers-Kronig invariant. In our previous investigation [31], anti-symmetric Lorentzians were successfully employed, whereas in this study, we utilized dielectric basis functions, which, according to the Lorentz model, are better suited for fitting absorption bands. The dielectric basis enabled excellent reconstructions of the complex permittivities for all six test substances. We demonstrated that using general, generic knowledge about locations and widths of typical organic substances as initial conditions for the reconstruction algorithm considerably enhanced the accuracy of the resulting reconstructions.

## Author contributions

**Conceptualization:** Proity Nayeeb Akbar, Reinhold Blumel.

**Data curation:** Proity Nayeeb Akbar.

**Formal analysis:** Proity Nayeeb Akbar.

**Investigation:** Proity Nayeeb Akbar.

**Methodology:** Proity Nayeeb Akbar, Reinhold Blumel.

**Project administration:** Reinhold Blumel.

**Resources:** Proity Nayeeb Akbar.

**Software:** Proity Nayeeb Akbar.

**Supervision:** Reinhold Blumel.

**Validation:** Proity Nayeeb Akbar, Reinhold Blumel.

**Visualization:** Proity Nayeeb Akbar.

**Writing – original draft:** Proity Nayeeb Akbar, Reinhold Blumel.

**Writing – review & editing:** Proity Nayeeb Akbar, Reinhold Blumel.

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
