## [Decision Letter · Decision Letter 0]

18 Dec 2024

PONE-D-24-54445Inverse Reconstruction of Model Cells: Extracting Structural and Molecular Insights through Infrared Spectroscopic CytologyPLOS ONE

Dear Dr. Akbar,

Thank you for submitting your manuscript to PLOS ONE. After careful consideration, we feel that it has merit but does not fully meet PLOS ONE’s publication criteria as it currently stands. Therefore, we invite you to submit a revised version of the manuscript that addresses the points raised during the review process.

We look forward to receiving your revised manuscript.

Kind regards,

Alemayehu Getahun Kumela, Ph.D.

Academic Editor

PLOS ONE

**Journal requirements:**

Reviewers' comments:

Reviewer's Responses to Questions

**Comments to the Author**

1. Is the manuscript technically sound, and do the data support the conclusions?

Reviewer #1: Yes

Reviewer #2: Yes

2. Has the statistical analysis been performed appropriately and rigorously? 

Reviewer #1: Yes

Reviewer #2: Yes

3. Have the authors made all data underlying the findings in their manuscript fully available?

Reviewer #1: Yes

Reviewer #2: Yes

4. Is the manuscript presented in an intelligible fashion and written in standard English?

Reviewer #1: Yes

Reviewer #2: Yes

5. Review Comments to the Author

**Reviewer #1: **At first, I want to declare that I am not professional in the area of complex permittivity measured by the infrared spectroscopy. This work by Proity Nayeeb Akbar and Reinhold Blümel describes the inverse reconstruction of complex permittivity through infrared spectroscopy. They also discussed the fits between simulated extinction efficiencies to the experimental extinction efficiencies. In total, they have compared the complex permittivity and extinction efficiencies of six materials named PMMA, PC, PDMS, PEI, PET and PS respectively. It was well written and sounds good. In view of such interesting results, Plos one is the potential journal to publish this work. Prior to further considerations, the following issues should be considered:

1. The authors listed the experimental permitivities (real and imaginary parts) and extinction spectrum together. However, I am not sure what are their relations between equation 3 and equations 10-11.

2. In the manuscript, the authors listed Figure 5, however, they did not discuss this figure in main text. Please check the 3.3 part dielectric functions versus anti-symmetrized lorentzians.

3. In the Figure 5 and 6, why the authors applied the constructed fitting of 17 optimized dielectric/anti-symmetric Lorentzian functions?

**Reviewer #2: **Well conceived manuscript. It would be good if recent studies ( references/ work done) would be added in the manuscript This study can add in the existing knowledge related infrared microspectroscopy. These findings are significant as they not only improve the accuracy of

the reconstructions, but also advance the use of IR microspectroscopy for application in clinical cytology,

where precise molecular analysis is vital in diagnosing and monitoring diseases at the cellular level.

6. PLOS authors have the option to publish the peer review history of their article (what does this mean?). If published, this will include your full peer review and any attached files.

Reviewer #1: **Yes: **Hong-Ying Gao

Reviewer #2: No

---

## [Author Response · Author response to Decision Letter 1]

28 Jan 2025

A rebuttal letter that responds to each point raised by the academic editor and reviewer(s) is upload. Please see the uploaded document named "Response to Reviewers."

---

## [Decision Letter · Decision Letter 1]

7 Feb 2025

PONE-D-24-54445R1Inverse Reconstruction of Model Cells: Extracting Structural and Molecular Insights through Infrared Spectroscopic CytologyPLOS ONE

Dear Dr. Akbar,

Thank you for submitting your manuscript to PLOS ONE. After careful consideration, we feel that it has merit but does not fully meet PLOS ONE’s publication criteria as it currently stands. Therefore, we invite you to submit a revised version of the manuscript that addresses the points raised during the review process.

**While you have addressed all reviewers concern in appropriate manner, you have to correct the Reference, It  is must be before full stop.**

Please submit your revised manuscript by  Mar 24 2025 11:59PM. If you will need more time than this to complete your revisions, please reply to this message or contact the journal office at plosone@plos.org. Please include the following items when submitting your revised manuscript:

We look forward to receiving your revised manuscript.

Kind regards,

Alemayehu Getahun Kumela, Ph.D.

Academic Editor

PLOS ONE

Journal Requirements:

Reviewers' comments:

Reviewer's Responses to Questions

**Comments to the Author**

1. If the authors have adequately addressed your comments raised in a previous round of review and you feel that this manuscript is now acceptable for publication, you may indicate that here to bypass the “Comments to the Author” section, enter your conflict of interest statement in the “Confidential to Editor” section, and submit your "Accept" recommendation.

Reviewer #1: All comments have been addressed

Reviewer #3: All comments have been addressed

2. Is the manuscript technically sound, and do the data support the conclusions?

Reviewer #1: Yes

Reviewer #3: Yes

3. Has the statistical analysis been performed appropriately and rigorously? 

Reviewer #1: Yes

Reviewer #3: Yes

4. Have the authors made all data underlying the findings in their manuscript fully available?

Reviewer #1: Yes

Reviewer #3: Yes

5. Is the manuscript presented in an intelligible fashion and written in standard English?

Reviewer #1: Yes

Reviewer #3: No

6. Review Comments to the Author

Reviewer #1: All my concerns have been properly answered, and the manuscript is visibly improved in quality. Thus, it is ready for publication.

Reviewer #3: Comments on manuscript PONE-D-24-54445R1, entitled "Inverse Reconstruction of Model Cells: Extracting Structural and Molecular Insights through Infrared Spectroscopic Cytology"

The study develops an advanced inverse scattering algorithm for accurately reconstructing the dielectric properties of single biological cells using infrared (IR) microspectroscopy. By analyzing spherical model cells filled with six organic test substances, the method successfully extracts complex permittivities, improving the accuracy of molecular analysis. The results show that using dielectric functions for reconstruction outperforms other methods, advancing the potential of IR microspectroscopy for precise cellular diagnostics, especially in detecting malignancies and monitoring diseases, with no prior knowledge of spectroscopic principles required. This technique holds promise for enhancing clinical cytology and disease diagnosis. Prior to publication, the following minor revisions are required:

1. The abstract is not as strong as it could be. While it includes critical components such as background, methodology, and significance, it can be improved for better clarity, readability, and focus.

2. According to PLOS ONE's submission guidelines, references should be cited in the text using square brackets, placed immediately after the relevant information and before any punctuation marks. Please check this.

3. What does "Opt" stand for?

4. Analysis and Interpretation: Figures 1-4 are not well interpreted and compared with the literature. Additionally, Figures 1-4 are not visible. Please make the necessary revisions.

5. The abstract needs revision.

7. PLOS authors have the option to publish the peer review history of their article (what does this mean?). If published, this will include your full peer review and any attached files.

Reviewer #1: **Yes: **Hong-Ying Gao

Reviewer #3: No

---

## [Author Response · Author response to Decision Letter 2]

18 Feb 2025

Please see the attached document "response letter to reviewers."

---

## [Decision Letter · Decision Letter 2]

23 Feb 2025

Inverse Reconstruction of Model Cells: Extracting Structural and Molecular Insights through Infrared Spectroscopic Cytology

PONE-D-24-54445R2

Dear Dr.  Proity Nayeeb Akbar,

We’re pleased to inform you that your manuscript has been judged scientifically suitable for publication and will be formally accepted for publication once it meets all outstanding technical requirements.

Kind regards,

Alemayehu Getahun Kumela, Ph.D.

Academic Editor

PLOS ONE

Reviewers' comments:

Reviewer's Responses to Questions

**Comments to the Author**

1. If the authors have adequately addressed your comments raised in a previous round of review and you feel that this manuscript is now acceptable for publication, you may indicate that here to bypass the “Comments to the Author” section, enter your conflict of interest statement in the “Confidential to Editor” section, and submit your "Accept" recommendation.

Reviewer #3: All comments have been addressed

2. Is the manuscript technically sound, and do the data support the conclusions?

Reviewer #3: Yes

3. Has the statistical analysis been performed appropriately and rigorously? 

Reviewer #3: Yes

4. Have the authors made all data underlying the findings in their manuscript fully available?

Reviewer #3: Yes

5. Is the manuscript presented in an intelligible fashion and written in standard English?

Reviewer #3: Yes

6. Review Comments to the Author

Reviewer #3: I have satisfied in authors response. The author has been adressed all my comments. Therefore I recommend accept it.

7. PLOS authors have the option to publish the peer review history of their article (what does this mean?). If published, this will include your full peer review and any attached files.

Reviewer #3: No

---

## [Editor Report · Acceptance letter]

PONE-D-24-54445R2

PLOS ONE

Dear Dr. Akbar,

I'm pleased to inform you that your manuscript has been deemed suitable for publication in PLOS ONE. Congratulations! Your manuscript is now being handed over to our production team.

Kind regards,

on behalf of

Dr. Alemayehu Getahun Kumela

Academic Editor

PLOS ONE